# The Diagnostic Value of Multimodal Contrast-Enhanced Ultrasound in Sentinel Lymph Nodes After Neoadjuvant Therapy for Breast Cancer

**DOI:** 10.3390/diagnostics15192432

**Published:** 2025-09-24

**Authors:** Jiaqian Zhong, Jia Luo, Jiaping Li, Manying Li, Yingli Liu, Jinyu Liang, Fushun Pan, Xiaoyan Xie, Yanling Zheng

**Affiliations:** Department of Medical Ultrasonics, Institute for Diagnostic and Interventional Ultrasound, The First Affiliated Hospital of Sun Yat-sen University, Guangzhou 510000, China; zhongjq29@mail2.sysu.edu.cn (J.Z.); luojia7@mail.sysu.edu.cn (J.L.); lijiap2@mail2.sysu.edu.cn (J.L.); limy87@mail.sysu.edu.cn (M.L.); liuyl33@mail3.sysu.edu.cn (Y.L.); ljyu@mail.sysu.edu.cn (J.L.); panfush@mail.sysu.edu.cn (F.P.); xiexyan@mail.sysu.edu.cn (X.X.)

**Keywords:** breast cancer, sentinel lymph node, neoadjuvant therapy, percutaneous contrast-enhanced ultrasound (PCEUS), intravenous contrast-enhanced ultrasound (IVCEUS)

## Abstract

**Objective:** Accurate diagnosis of sentinel lymph node (SLN) status after neoadjuvant therapy (NAT) for breast cancer is crucial for guiding axillary management. This study aimed to evaluate novel contrast-enhanced ultrasound (CEUS) patterns for assessing SLNs following NAT. **Methods:** We retrospectively analyzed clinical and imaging data from 279 breast cancer patients who completed NAT and underwent surgery between June 2019 and December 2024. Preoperative SLN evaluations included percutaneous CEUS (PCEUS), intravenous CEUS (IVCEUS), and conventional ultrasound (CUS). Intraoperative SLN biopsy was performed using methylene blue tracer, with pathological results serving as the gold standard. Diagnostic efficacy was compared among CUS, previously used PCEUS patterns, newly proposed PCEUS, IVCEUS, and combined CEUS. **Results**: The newly proposed PCEUS classified SLNs into six types, while IVCEUS categorized enhancement into three sequences and four patterns. Among the 347 SLNs detected via PCEUS, 292 (84.15%) were benign and 55 (15.85%) were malignant. The newly proposed PCEUS demonstrated higher diagnostic efficacy compared to CUS, prior PCEUS patterns, IVCEUS, and combined CEUS, with sensitivity, specificity, positive predictive value, negative predictive value, accuracy, and area under the curve of 49.1% (27/55), 86.3% (252/292), 40.3% (27/67), 90.0% (252/280), 80.4% (279/347), and 0.677 (95% CI: 0.625–0.726), respectively. However, DeLong tests revealed no statistically significant differences between the methods (all *p* > 0.05). **Conclusions**: The novel CEUS classification improved diagnostic accuracy for SLNs after NAT, though accuracy remains relatively low. Future integration of artificial intelligence may further enhance diagnostic efficacy.

## 1. Introduction

Breast cancer is the most common malignant tumor in women, exhibiting the highest incidence among all female malignancies [1]. Neoadjuvant therapy (NAT) prior to surgery has become a critical component of multimodal breast cancer management, with long-term survival outcomes comparable to those of postoperative adjuvant chemotherapy [2]. In locally advanced breast cancer, NAT reduces primary tumor burden and downstages axillary lymph node metastases, facilitating breast-conserving surgery and minimizing the extent of axillary dissection. This is particularly advantageous compared to complete axillary lymph node dissection (ALND), which carries risks of complications such as shoulder stiffness, arm lymphedema, numbness, paresthesia, chronic pain, limited mobility, lymphangitis, and tissue fibrosis, thereby adversely affecting patients’ quality of life. Approximately 35–68% of lymph node-positive breast cancer patients achieve axillary pathological complete response after NAT [3]. According to the National Surgical Adjuvant Breast and Bowel Project (NSABP) B32 trial [4], patients with negative SLNs showed similar regional recurrence rates, disease-free survival, and overall survival following SLN biopsy (SLNB) or ALND, though SLNB is associated with significantly fewer postoperative adverse effects. For these patients undergoing NAT, the accuracy of post-NAT SLNB remains acceptable, with a random-effects meta-analysis estimating an overall false negative rate (FNR) of 7% (95% CI: 5–9%) [5]. However, earlier studies reported FNRs ranging from 8% to 40% for SLNB after NAT in clinically node-positive (cN+) patients [6,7,8]. International guidelines recommend maintaining an SLNB FNR below 10% [9], as meeting this threshold ensures that false-negative events do not adversely affect postoperative tumor recurrence or survival outcomes. Data on the long-term oncology outcomes of this less invasive approach remains limited, and axillary surgical guideline recommendations for CN+ patients still vary widely [10]. In this context, assessing post-NAT sentinel lymph nodes is critical to planning the most appropriate surgical staging method. Therefore, optimizing the diagnostic accuracy of SLN evaluation after NAT necessitates a multimodal precision approach.

Ultrasound is the preferred imaging modality for evaluating axillary lymph nodes in breast cancer patients after NAT [11]. However, for patients with initial lymph node involvement, conventional ultrasound has limitations in determining axillary status post-NAT [12] probably due to post-NAT fibrosis and tumor regression complicating ALN evaluation. Moreover, its value in localizing and qualitatively assessing SLNs is limited [13]. Contrast-enhanced ultrasound (CEUS) is an emerging technique that utilizes microbubbles smaller than red blood cells, enabling capillary and lymphatic penetration. Percutaneous CEUS (PCEUS) involves the subcutaneous or lymphatic injection of a contrast agent, which is absorbed by the lymphatic system, allowing real-time imaging of lymphatic vessels and nodes [14]. However, the diagnostic efficacy of PCEUS varies across studies [15,16]. In addition to PCEUS, there is another technique, intravenous CEUS (IVCEUS) that involves the intravenous injection of a contrast agent followed by real-time monitoring to evaluate SLN location, condition, and blood perfusion, aiding in the detection of local defects or perfusion abnormalities. One study [17] reported AUC values of 0.906 (95% CI: 82.2–99.0%) for lymphatic CEUS (LCEUS) and 0.872 (95% CI: 81.4–98.2%) for intravenous CEUS (ICEUS), with no significant difference between the methods (*p* = 0.752). Combined CEUS was highly effective but not superior to LCEUS alone, possibly due to a small sample size and simplified ICEUS diagnostic criteria. A subsequent study [18] refined PCEUS and IVCEUS classifications. The proposed PCEUS classification method combined with the structure of lymph nodes, and the IVCEUS method combined with the enhancement order and enhancement type. The results showed that among the previously classified PCEUS [19], newly classified PCEUS, IVCEUS, and combined newly classified CEUS, the combined newly classified CEUS had the highest diagnostic efficacy. This indicates that the newly classified CEUS patterns have better diagnostic efficacy for SLNs in early breast cancer.

To our knowledge, no prior study has applied CEUS to evaluate SLNs after NAT for breast cancer. Therefore, the primary objective of this study is to employ the novel classification to assess SLN status in breast cancer patients following NAT.

## 2. Materials and Methods

### 2.1. Study Population

The study was conducted in accordance with the Declaration of Helsinki (as revised in 2013) and approved by the Ethics Committee of the First Affiliated Hospital of Sun Yat-sen University (Approval No. [(2020)316], Date: [2 September 2020]). Informed consent was obtained from all the patients. Patients with breast cancer who received NAT at the First Affiliated Hospital of Sun Yat-sen University between June 2019 and December 2024 were enrolled based on the inclusion and exclusion criteria. The inclusion criteria were as follows: (1) histopathologically confirmed breast cancer before NAT; (2) complete clinical and pathological data; (3) completion of the entire NAT course and subsequent breast and axillary surgery at our hospital; (4) postoperative axillary SLNB pathology or ALND result being completely negative; (5) preoperative complete SLN CEUS examination (both PCEUS and IVCEUS). The exclusion criteria included the following: (1) bilateral breast cancer lesions, which may interfere with the ultrasound accuracy of unilateral sentinel lymph node assessment; (2) distant metastasis at initial diagnosis; (3) poor ultrasound image quality; (4) non-visualization of SLNs or lymphatic vessels during PCEUS; (5) patients without an axillary surgery plan. Ultimately, 279 patients with 347 SLNs were included (Figure 1).

### 2.2. Ultrasound Examination and Image Analysis

Within approximately two weeks before surgery, patients underwent breast and axillary ultrasound examinations using Siemens ACUSON (Siemens Healthineers, Erlangen, Germany; 10L4 probe), Mindray Resona 7 (Mindray, Shenzhen, China, L9-3 probe), Mindray Resona 9 (L9-3U probe), Philips iU22 (Philips, Amsterdam, The Netherlands; L9-3 probe), or Canon Aplio1900 (Canon Medical Systems, Tokyo, Japan; 18LX5 probe) systems. Continuous real-time CEUS imaging was performed under a low mechanical index (MI < 0.15). Board-certified radiologists conducted breast and ALN ultrasound examinations in accordance with the Breast Imaging Reporting and Data System lexicon. During the ultrasound examination, patients were positioned supine with both arms raised above the head and externally rotated to fully expose the breast and axilla. The largest lesion section was selected for CEUS examination. Patients were instructed to breathe calmly during the procedure, and the ultrasound contrast agent was a suspension prepared by SonoVue (Bracco, Italy) lyophilized powder (59 mg sulfur hexafluoride microbubbles) mixed with 5 mL normal saline, which had been shaken for 30 s for standby. Then, 1 mL of the suspension was injected subcutaneously in the outer upper quadrant of the areola on the affected side, followed by gentle massage to promote lymphatic flow. SLN enhancement patterns were dynamically observed in dual-image mode and lymphatic drainage directions and SLN surface projections were marked. Finally, the probe was positioned at the surface projection of the SLN, switched to two-dimensional mode, and the largest section was selected. A total of 2.4 mL of microbubble suspension was injected intravenously to assess perfusion while keeping the probe stationary.

Subsequently, ultrasound characteristics of the SLN were documented, including cortical thickness, fatty hilum, PCEUS enhancement type, and sequence and type of IVCEUS. All ultrasound features were reviewed and recorded by three radiologists with at least five years of experience in breast ultrasound. Any discrepancies were resolved through consultation and consensus.

### 2.3. CUS and CEUS Classification of SLN

According to the cortical morphological features, each SLN was divided into one of the following four types [20]: Type I, thin and uniform hypoechoic cortex; Type II, uniform hypoechoic cortex, cortex thickness is less than 3 mm; Type III, focal cortex thickening or uneven thickening, the thickness is greater than or equal to 3.0 mm; and Type IV, totally hypoechoic node with no hyperechoic hilum. Type III–IV were diagnosed as malignant lymph nodes, and Types I–II as benign lymph nodes.

In this study, different from the previous classification of PCEUS, the SLN-PCEUS enhancement patterns were divided into six types (eight subtypes) (Table 1): type I showed only part of the cortex was enhanced, others were non-enhanced, and the lymph node cortex was unevenly thickened; type II showed a partial cortical filling defect; type III showed non-enhancement; type IV showed a homogeneous high enhancement; type V showed diffuse inhomogeneous high enhancement; type VI included type VIa which showed non/low enhancement of the lymphatic hilus, homogeneous high enhancement of the cortex, type VIb of which one half showed Type IV, V or VIa, and the other showed non-enhancement, and type VIc which showed that only part of the cortex was enhanced, others were not enhanced, and the lymph node cortex was evenly thickened. Types I–III were diagnosed as malignant lymph nodes, and Types IV–VI as benign lymph nodes.

In addition, the SLN-IVCEUS enhancement patterns were divided into four types: type I showed homogeneous high enhancement; type II showed diffuse inhomogeneous high enhancement; type III showed no/low enhancement of the lymphatic hilus, homogeneous high enhancement of the cortex; type IV showed part of the cortical filling defect, low enhancement or high enhancement, the rest showed the performance of types I, II or III. And SLN-IVCEUS enhancement sequence was divided into three types based on the orders of bubbles entering the lymph nodes: centrifugal enhancement, centripetal enhancement, and diffuse enhancement. A malignant lymph node was diagnosed as long as one of the following conditions was met: (I) centripetal enhancement; (II) diffuse enhancement; (III) the enhancement mode of IVCEUS was type IV. SLN was diagnosed as malignant when PCEUS and/or IVCEUS diagnosed the SLN as malignant (Figure 2 and Figure 3).

### 2.4. Clinical and Pathological Assessment

Before NAT, patients’ assessments included age, tumor size, axillary lymph node status, estrogen receptor (ER), progesterone receptor (PR), human epidermal growth factor receptor 2 (HER-2), and Ki-67 status. ER, PR, HER-2, and Ki-67 statuses were evaluated via the immunohistochemical analysis of fine needle aspiration samples obtained before NAT. Hormone receptor positive was identified as ≥1% of nuclear staining for ER or PR. HER-2 status was classified as 0, 1+, 2+, or 3+. A HER-2 staining intensity score of 3+ was considered positive, while 0 or 1+ was considered negative. Additionally, tumors scored 2+ underwent fluorescence in situ hybridization for HER-2 determination.

### 2.5. Surgical Management of the ALN

All patients underwent breast surgery and axilla-related surgery (SLNB or SLNB plus ALND or ALND) after NAT. SLNB was guided by blue dye mapping. If no blue-dyed SLN was visualized, the PCEUS-marked SLNs were excised. All marked SLNs by blue dye or PCEUS were removed and sent for frozen and paraffin pathological examination, and ALND was performed immediately if intraoperative pathology confirmed SLN positivity. Each axillary lymph node was stained with hematoxylin and eosin to detect malignant cells and residual lesions. The number of lymph nodes with and without metastasis was recorded. Negative SLN status was defined as negative SLNB result or negative result from ALND alone. According to the AJCC Cancer Staging Manual (8th edition), LNs containing isolated tumor cells (ITCs), micrometastases, or macrometastases were classified as positive. The procedure was performed by two surgeons with extensive surgical experience.

### 2.6. Data Analysis

SPSS 25.0 and MedCalc 20.1.0 were used for statistical analysis. Data were summarized using standard descriptive statistics and frequency tables. SLN status was based on pathological results. The area under the receiver operating characteristic curve (AUC), sensitivity, specificity, positive predictive value, negative predictive value, and accuracy were calculated. Diagnostic efficacy of different examination methods was compared with the DeLong test. A two-sided *p* < 0.05 was considered as a statistically significant difference.

## 3. Results

### 3.1. Patient and Tumor Characteristics

In this study, 279 female breast cancer patients were included. The mean participant age was 48.9 ± 10.3 years (range: 25–79 years). Clinical characteristics of these patients are summarized in Table 2.

Imaging findings of lymphatic channels and sentinel lymph nodes visualized by PCEUS after NAT are summarized in Table 3. In 74 patients, no blue-dyed SLN was observed intraoperatively; in these cases, SLNs were excised based on PCEUS markings.

### 3.2. Correlation Between SLN Morphological Characteristics and CEUS Characteristics

Among SLNs classified as benign by combined CEUS, the majority (186/239, 77.8%) exhibited normal grayscale ultrasound morphological features, including preserved nodal architecture, thin cortical thickness (<3 mm), and regular hilar structure. Paradoxically, 62.0% (67/108) of CEUS-identified malignant nodes displayed benign grayscale morphological characteristics. This discordance suggests that while macroscopic tumor regression may restore cortical morphology, residual micrometastases could still generate detectable CEUS enhancement patterns.

### 3.3. Diagnostic Efficacy of Different Diagnostic Methods in SLNs

Of the 347 SLNs examined, 55 (15.9%) demonstrated malignant pathology, while 292 (84.1%) were benign. Notably, skip metastases were observed in 31 non-SLNs from 14 patients despite negative SLNs identified by tracer localization techniques (PCEUS or blue dye). Pathological characteristics and CEUS findings of SLNs are detailed in Table 4.

Comparative analysis of PCEUS and IVCEUS for SLN evaluation is summarized in Table 5. The McNemar test indicated significant diagnostic consistency between the two modalities (*p* = 0.009, *p*< 0.001), demonstrating strong agreement in metastatic status assessment.

Among 279 axillary regions evaluated, 230 (82.4%) showed no nodal metastasis. Preoperative imaging accurately identified 170 nodal regions (73.9%) as metastasis-negative, suggesting that integrated PCEUS and IVCEUS evaluation may help stratify patients for SLNB to some extent, thereby reducing unnecessary ALND in clinically node-negative cases.

The diagnostic performance of different US characteristics is presented in Table 6. Accuracies of conventional US, the previous classification of PCEUS, new PCEUS, IVCEUS and combined CEUS were 72.9%, 48.7%, 80.4%, 73.2%, and 72%, respectively. The ROC curves of different diagnostic methods are shown in Figure 4. Although the novel PCEUS classification demonstrated a higher diagnostic AUC value compared to the conventional classification system (0.677 vs. 0.617), DeLong’s test revealed no statistically significant differences in diagnostic efficacy between the pairwise comparisons of the two methodologies. Furthermore, the novel PCEUS classification showed comparable AUC value to the combined ultrasound modality, suggesting IVCEUS may not yield substantial incremental diagnostic value in this clinical context.

## 4. Discussion

Due to the therapeutic effects of neoadjuvant systematic therapy, initial cN+ patients can achieve an axillary pathologic complete response and are thought not to benefit from ALND [21]. Less invasive restaging via SLNB aims to reduce related-ALND surgical morbidity but it is worth highlighting that intraoperative frozen-section analysis of SLNs has limitations in the NAT setting, with a reported sensitivity of less than 80% for micrometastases and ITCs [22]. Therefore, this may be one of the reasons for the FNR of SLNB exceeding the acceptable range. The elevated FNR associated with post-NAT SLNB highlights the inherent challenges and clinical risks of relying solely on this modality for breast cancer staging. Our cohort showed an 80% ypN0 rate among cN+ breast cancer patients following NAT, indicating a substantial proportion may be candidates for ALND omission. Notably, elevated ypN0 rates were particularly pronounced in tumors exhibiting high Ki67 proliferative indices, suggesting the biological profile may predict enhanced therapeutic responsiveness to NAT and identify candidates for de-escalated axillary strategy. These findings underscore the critical need for developing advanced multimodal imaging protocols or molecular profiling techniques to predict axillary pathologic response preoperatively, thereby facilitating precise therapeutic decision-making. In order to accurately determine the status of SLNs, our study adopted a non-invasive CEUS. In this study, the intraoperative frozen section of a patient showed no SLN metastasis (0/3), but the subsequent paraffin section showed ITC in one and macrometastases in two nodes. In this patient, we evaluated the status of SLN as metastatic before surgery, suggesting that CEUS can provide certain reference value and lay a certain foundation for clinicians to decide whether supplementary axillary lymph node dissection is necessary.

PCEUS aids in both SLN anatomical localization and diagnosis. Prior studies have stratified SLN PCEUS enhancement patterns into three distinct categories which includes homogeneous enhancement, heterogeneous enhancement, and no enhancement, with substantial heterogeneity observed in diagnostic efficacy [15,23,24]. Homogeneous enhancement of SLN was considered benign, while heterogeneous and no enhancement indicated malignancy. In this study, we used a more complete classification modality that has recently been proposed, and it performs well in the SLN diagnosis of early breast cancer. The PCEUS enhancement patterns of SLNs were systematically integrated with their structural characteristics, leading to the establishment of a novel six-type classification system. We found that 92% of homogeneously enhanced SLNs were benign, and 35% of non-enhanced SLNs were malignant. Interestingly, the non-enhancement patterns observed in these cases deviated significantly from prior diagnostic background. Then a retrospective analysis of IVCEUS findings revealed most of these SLNs displayed heterogeneous high enhancement or low enhancement at the lymphatic hilum and others expressed homogeneous enhancement. We hypothesize that this result may be attributed to chemotherapy-induced regression of intranodal lesions, which alters lymphatic drainage pathways. During PCEUS, contrast agents fail to penetrate the fibrotic zones of regressed lymph nodes, manifesting as non-enhancement patterns. Conversely, IVCEUS demonstrates heterogeneous enhancement: preserved normal parenchyma exhibits moderate hyperenhancement, while post-chemotherapy fibrotic regions display hypoenhancement. This spatial discordance likely reflects histopathological transformations such as tumor necrosis and stromal fibrosis, creating compartmentalized perfusion barriers so that contrast agents distribute differently.

Heterogeneously enhanced SLNs were subdivided into types I–II (malignant) and types V–VI (benign). Pathology confirmed malignancy in 50.0% of Type I and 40.0% of Type II SLNs. The attenuated diagnostic efficacy observed in this cohort may be partially attributable to the low prevalence of these types (4.3% and 11.2%, respectively). This limited representation of characteristic SLN features could introduce sampling bias, thereby reducing the statistical power to detect significant associations between imaging patterns and pathological outcomes. Otherwise, 84.9% of Type V, 97.6% of Type VIa, 85.7% of Type Vib, and 81.2% of Type VIc SLNs were pathologically benign. This categorization may relate to lymphatic reflux or node structure. As reported in one study [25], partial enhanced SLN showed certain histopathological characteristics. It demonstrated either expansion of lymphoid follicles or distension of the lymphatic sinuses, whereas fully enhanced SLNs maintained normal lymphoid architecture. The stratification of heterogeneous enhancement patterns via PCEUS may refine diagnostic precision in characterizing SLN pathology. Overall, the new PCEUS classification showed higher specificity (86.3% vs. 66.1%) and accuracy (80.4% vs. 48.7%) than the previous method, though differences were not statistically significant. Notably, the observed discrepancies in diagnostic performance between the novel classification system and the conventional three-type framework could be potentially attributable to cohort and methodological heterogeneity.

PCEUS could reflect the lymphatic drainage of lymph nodes, while IVCEUS could reflect the blood supply. This investigation adopted a novel IVCEUS classification system based on enhancement order and patterns for SLNs, diverging from prior research paradigms. Malignant lymph nodes exhibited non-centrifugal contrast perfusion sequences, attributed to tumor-induced neovascularization disrupting physiological hemodynamics. Histopathological correlation revealed that cortical enhancement heterogeneity in these nodes (Type IV pattern) stemmed from architecturally discordant microvascular networks between metastatic deposits and native lymphoid stroma, creating perfusion discontinuities observable on contrast kinetics. But this manifestation occurred in only nine SLNs, of which 33.3% were malignant. This may be due to the effect of NAT on tumor vasculature. Several studies [26,27,28] have shown that effective neoadjuvant chemotherapy administration impacts tumoral angiogenic activity. After combining the newly classified PCEUS and IVCEUS, it was found that the specificity and positive predictive value were improved compared to the three-classified PCEUS, but there was no significant difference. Not only that, compared with the new classified PCEUS alone, the combined CEUS only improved sensitivity and negative predictive value, while other performances did not show improvement. This indicates that the application effect of this method in the judgment of SLN after NAT differs from that in the judgment of SLN in early breast cancer, and there are many innate NAT-induced factors complicating our naked-eye judgment for SLNs.

We acknowledge several limitations in our study. First, the retrospective single-center study may limit generalizability. And large prospective validations of this method are needed in the future. Second, the exclusion of patients with no SLNs and SLCs detected by CEUS may introduce selection bias. Thirdly, only CEUS was evaluated in this study; future studies could include super-resolution ultrasound (SRUS), which has shown promise in predicting metastatic SLNs [29]. Fourthly, the small tumor burden including micrometastases and ITCs in SLNs could contribute to the lack of significant differences in CEUS enhancement pattern between tumor-free and metastatic SLNs. Additionally, post-treatment changes such as fibrosis, calcification, mucin pools, and foamy histiocyte aggregates may be misdiagnosed as malignant on PCEUS. Finally, although the new CEUS method has improved the diagnostic efficiency compared with the previous classification method, the diagnostic accuracy is still less than 85%. Thus, it is necessary to integrate artificial intelligence (AI)-driven image analysis to capture subvisual features which may enhance diagnostic precision and address current limitations of human interpretation, thereby improving the diagnosis of SLN status after NAT in the future.

## 5. Conclusions

The newly classified CEUS approach offers a non-invasive method for SLN assessment post-NAT but requires further technological refinement for clinical utility. AI integration may optimize diagnostic precision by overcoming limitations of human interpretation.

## Figures and Tables

**Figure 1 diagnostics-15-02432-f001:**
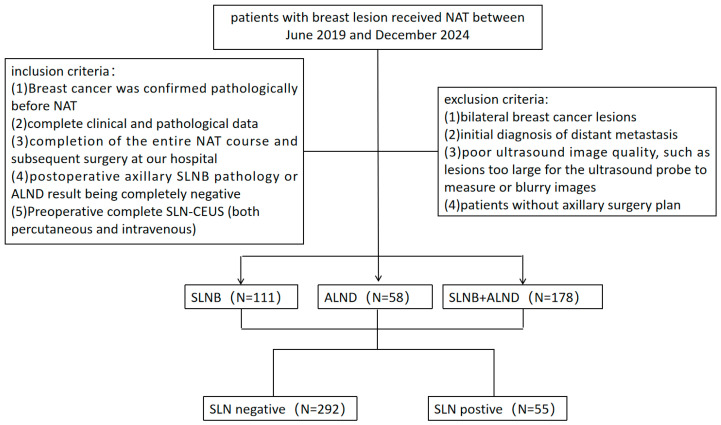
Flowchart of the study population.

**Figure 2 diagnostics-15-02432-f002:**
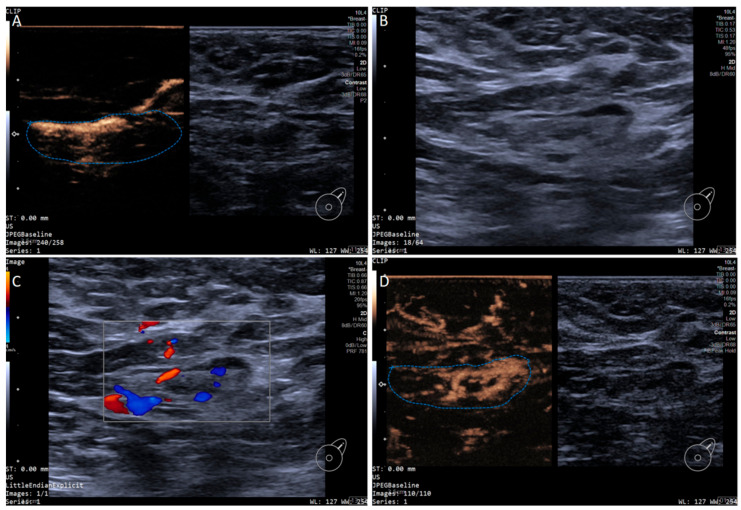
Multimodal ultrasound imaging of pathologically confirmed non-metastatic sentinel lymph node. (**A**) Intradermal CEUS image (the enhancement pattern is type VIc, the blue dotted line delineates the contour of the SLN); (**B**) 2D US image; (**C**) Color Doppler Flow Imaging (CDFI) image; (**D**) intravenous CEUS image (the enhancement pattern is III, the blue dotted line delineates the contour of the SLN).

**Figure 3 diagnostics-15-02432-f003:**
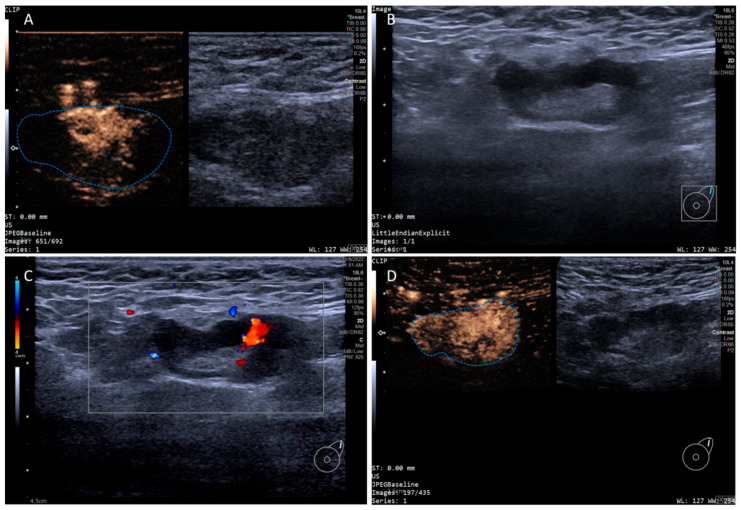
Multimodal ultrasound imaging of pathologically confirmed metastatic sentinel lymph node. (**A**) Intradermal CEUS image (the enhancement pattern is type I, the blue dotted line delineates the contour of the SLN); (**B**) 2D US image; (**C**) CDFI image; (**D**) intravenous CEUS image (the enhancement pattern is II, the blue dotted line delineates the contour of the SLN).

**Figure 4 diagnostics-15-02432-f004:**
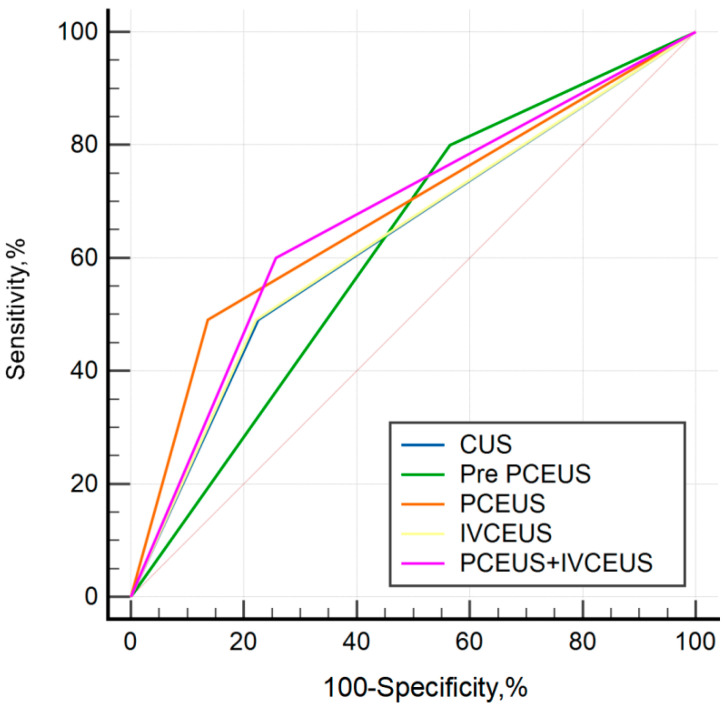
ROC curves of different diagnostic methods.

**Table 1 diagnostics-15-02432-t001:** The summary of previous and new SLN PCEUS classifications.

Method	Previous Classification of PCEUS	New Classification of PCEUS
Benign	type I, homogeneous enhancement	type IV, homogeneous high enhancement
	type II, inhomogeneous enhancement	type VIa, non/low enhancement of lymphatic hilus and homogeneous high enhancement of cortex
		type VIb, one half showed Type IV, V or VIa, and the other showed non-enhancement
		type VIc, only part of the cortex was enhanced, others were not enhanced, and lymph node cortex was evenly thickened
Malignant	type III, no enhancement	type I, only part of the cortex was enhanced, others were non-enhanced, and the lymph node cortex was unevenly thickened
		type II, partial cortical filling defect
		type III, non-enhancement

**Table 2 diagnostics-15-02432-t002:** Participant characteristics.

Parameter	Value
Age (years)	48.9 ± 10.3
Histologic type	
invasive ductal carcinoma	309
invasive lobular carcinoma	6
Others	29
Molecular subtype of breast tumor	
Luminal A	19
Lumnial B HER2−	96
Lumnial B HER2+	101
HER2+	72
TNBC	59
ALN status before NAT	
negative	72
positive	275
Pathological acquisition of lymph nodes	
puncture	283
imaging examination	64

**Table 3 diagnostics-15-02432-t003:** The identification of SLNs and SLCs by PCEUS.

Parameter	Value
Sentinel lymphatic channel	
1	220
2	108
3	11
>3	8
SLN number	
1	218
2	105
3	24

**Table 4 diagnostics-15-02432-t004:** US and pathology of SLN.

Diagnostic Method	Pathology		Total
	Benign	Malignant	
CUS			
I	209	23	232
II	17	5	22
III	64	24	88
IV	2	3	5
PCEUS			
I	6	6	12
II	21	14	35
III	13	7	20
IV	127	11	138
V	45	8	53
VIa	42	1	43
VIb	12	2	14
VIc	26	6	32
IVCEUS enhancement sequence			
Centrifugal	229	30	259
Centripetal	9	2	11
Diffuse	54	23	77
IVCEUS enhancement pattern			
I	176	27	203
II	81	21	102
III	29	4	33
IV	6	3	9
IVCEUS diagnosis			
benign	227	28	255
malignant	65	27	92
Combined CEUS			
benign	217	22	239
malignant	75	33	108
Total	292	55	347

**Table 5 diagnostics-15-02432-t005:** PCEUS and IVCEUS diagnosis of SLN.

PCEUS	IVCEUS		Total
	Benign	Malignant	
benign	225	55	280
malignant	30	37	67
Total	255	92	347

**Table 6 diagnostics-15-02432-t006:** Comparison of diagnostic efficacy of different diagnostic methods for SLN.

Diagnostic Method	Sensitivity	Specificity	Positive Predictive Value	Negative Predictive Value	Accuracy	AUC (95% CI)
CUS	49.10%	77.40%	89.00%	29.00%	72.90%	0.632 (0.579–0.683)
Previous classification of PCEUS	76.40%	66.10%	20.10%	92.00%	48.70%	0.617 (0.564–0.669)
PCEUS	49.10%	86.30%	40.30%	90.00%	80.40%	0.677 (0.625–0.726)
IVCEUS	49.15%	77.70%	29.30%	89.00%	73.20%	0.634 (0.581–0.685)
Combined CEUS	60%	74.30%	30.60%	90.80%	72.00%	0.672 (0.619–0.721)

Accuracy = (TP + TN)/(TP + TN + FP + FN); TP: true positive, TN: true negative, FP: false positive; FN: false negative.

## Data Availability

The data are not publicly available due to containing information that could compromise the privacy of research participants.

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
