# Peer review of "The Diagnostic Value of Multimodal Contrast-Enhanced Ultrasound in Sentinel Lymph Nodes After Neoadjuvant Therapy for Breast Cancer"

_diagnostics, 2025, doi:10.3390/diagnostics15192432_

Round 1
Reviewer 1 Report
Comments and Suggestions for Authors The article :"The diagnostic value of multimodal contrast-enhanced ultrasound in sentinel lymph nodes after neoadjuvant therapy for breast cancer" is of current interest in the management of the axilla in patients with breast gland cancer after neoadjuvant treatment. PCEUS demonstrates dual utility in both guiding their anatomical localization and
diagnosing SLN status.
The study is valuable as it includes a large number of patients, describing the clinical, pathological, and molecular characteristics. During PCEUS, contrast agents fail to penetrate the fibrotic peripheral areas of the regressed lymph nodes, manifesting as non-enhancement patterns. This investigation adopted a novel IVCEUS classification system delineating enhancement order and patterns for SLNs, diverging from prior research paradigms .After combining the newly classified PCEUS and IVCEUS, it was found that the specificity and positive predictive value were improved compared to the three-classified PCEUS, even so there was
no significant difference. I appreciate the comparative statistical analysis of PCEUS and IVCEUS for SLN evaluation described in table 4.
The work is valuable for its approach to introducing CEUS in the evaluation of the status of sentinel axillary lymph nodes after neoadjuvant chemotherapy and proposes new perspectives for evaluation using AI. The limitations of the study are honestly stated and I believe that the major downside of this research is its retrospective nature.
Nevertheless, I congratulate the group of authors and suggest continuing the research with a prospective cohort.
Author Response
Comment: The work is valuable for its approach to introducing CEUS in the evaluation of the status of sentinel axillary lymph nodes after neoadjuvant chemotherapy and proposes new perspectives for evaluation using AI. The limitations of the study are honestly stated and I believe that the major downside of this research is its retrospective nature.
Response: Thank you very much for the positive and encouraging feedback. We will accept this suggestion to conduct a prospective validation study next.
Reviewer 2 Report
Comments and Suggestions for Authors
Dear Editor-In-Chief
Diagnostic MDPI,
Subject: Review of the article diagnostic-3821962
Entitled “The diagnostic value of multimodal contrast-enhanced ultrasound in sentinel lymph nodes after neoadjuvant therapy for breast cancer”
This study aims to evaluate the diagnosis accuracy of SLN after NAT using CEUS. The study is relevant for the readers of this Journal. The introduction and methodology are described in necessary details. Some figures are not informative, some tables appear with poor quality and few acronyms should be spelled at first appearance. Some general comments are related to the journal formatting requirements, objective/ aim and imaging fundamentals. Below are some general and specific comments for the authors to consider.
General comments
- According to the journal formatting requirements, the abstract should not be structed i.e., no sections.
- Citations needs amendment across the entire manuscript, square brackets need to be used. Example regimens 2 should be regimens [2], NAT3 should be NAT [3] etc.
- While the pre-assessment of NAT is clear, what is the value of assessment of SLN post NAT?
- The phrase benign grayscale characteristic implies that there is a cutoff value to distinguish benign from malignant relaying on pixel value/ pixel depth value. Elaborate with supporting evidence.
Specific comments:
- line 47: NSABP is an acronym that needs to be spelled-out in the first presence.
- lines 82: the highest diagnostic efficacy can’t be measured by significance! This needs to be corrected.
- fig 1: should be presented in the correct order i.e. should be displayed before fig 2.
- line 136: it is recommended to present the summarize old and new SLN classification in a table.
- line 168: CDFI is an acronym that needs to be spelled out in the first presence.
- line 180: what is HR? Similarly, line 181: what is IHC?
- table 2: is with poor quality, showing as two sets of data were superimposed. The last 3 rows in tables 3 are facing the same.
- table 5: how was the accuracy calculated?
- line 31 (abstract): the novel CEUS classification improved diagnostic accuracy of SLNs post-NAT but lacked statistical superiority. This is quite confusing, call for rephrasing and re-writing.
Author Response
We sincerely thank you for the thorough and constructive comments, which have greatly helped us improve the quality of our manuscript. Below we provide a point-by-point response to each comment.
General comments:
Comment 1: According to the journal formatting requirements, the abstract should not be structed i.e., no sections.
Response 1: Thanks for your reminder. We have revised the abstract to remove the section headings (e.g.,Objective, Methods, Results, Conclusion) to comply with the journal’s format.
Comment: 2. Citations needs amendment across the entire manuscript, square brackets need to be used. Example regimens 2 shouldbe regimens [2], NAT3 should be NAT [3] etc.
Response 2: Thank you for your advice. We have revised all citations to the correct format, e.g., “regimens [2]”, “NAT [3]”.
Comment 3: While the pre-assessment of NAT is clear, what is the value of assessment of SLN post NAT?
Response 3: We have added a clarification in the Introduction section regarding the clinical relevance of SLN assessment after NAT, because part of these metastasis SLN can achieve PCR. These patients can avoid unnecessary axillary lymph node dissection.
Comment 4: The phrase benign grayscale characteristic implies that there is a cutoff value to distinguish benign from malignant relaying on pixel value/pixel depth value. Elaborate with supporting evidence.
Response 4: Thank you for your advice. We have revised the relevant section to clarify that the term refers to morphological features (e.g.,cortical thickness, hilar structure, shape) rather than pixel-based measurements.
Specific Comments:
1. line 47: NSABP is an acronym that needs to be spelled-out in the first presence.
Response: Thank you for your advice. “National Surgical Adjuvant Breast and Bowel Project (NSABP)” has been added.
2. lines 74: the highest diagnostic eficacy can't be measured by significance! This needs to be corrected.
Response: Thank you for your reminder. We have rephrased the sentence to clarify that statistical significance was used to compare diagnostic performance, not to measure efficacy directly.
3. fig 1: should be presented in the correct order i.e. should be displayed before fig
Response:Thank you for your advice. The figures have been reordered accordingly.
4. Line 136: it is recommended to present the summarize old and new SLN classification in a table.
Response: Thank you for your advice. A new table (Table 2) has been added to compare the previous and revised classification systems.
5. line 168: CDFl is an acronym that needs to be spelled out in the first presence.
Response: Thank you for your advice. “Color Doppler Flow Imaging (CDFI)” has been added.
6. line 180: what is HR? Similary, line 181: what is lHC?
Response: Thank you for your advice. “Hormone receptor (HR)” and “Immunohistochemistry (IHC)” have been spelled out.
7. table 2. is with poor quality, showing as two sets of data were superimposed. The last 3 rows in tables 3 are facing the same.
Response: Thank you for your advice. Both tables have been reformatted for clarity and accuracy.
8. table 5: how was the accuracy calculated?
Response: Thank you for your quest. We have added a footnote to Table 5 explaining the formula: Accuracy = (TP + TN) / (TP + TN + FP + FN).
9. line 31 (abstract): the novel CEUS classification improved diagnostic accuracy of SLNs post-NAT but lacked statistical superiority. This is quite confusing, call for rephrasing and re-writing.
Response: Thank you for your advice. The sentence has been revised for clarity: “the novel CEUS classification improved diagnostic accuracy for SLNs after NAT, though accuracy remains relatively low.”